Establishment and characterization of fantail goldfish fin (FtGF) cell line from goldfish, Carassius auratus for in vitro propagation of Cyprinid herpes virus-2 (CyHV-2)

Dharmaratnam Arathi 1
Kumar Raj 1
Valaparambil Basheer Saidmuhammed 1
Sood Neeraj 2
Pradhan Pravata Kumar 2
Das Sweta 1
Swaminathan T. Raja 1 Thangaraj.swaminathan@icar.gov.in
1 Peninsular and Marine Fish Genetic Resources Centre, ICAR National Bureau of Fish Genetic Resources , Kochi, Kerala , India
2 ICAR National Bureau of Fish Genetic Resources , Lucknow, Uttar Pradesh , India
Esteban María Ángeles
Electronic publication date: 2020 Jul 14
Publication date: 2020
Volume: 8
Electronic Location ID: e9373
Received 2019 Dec 18; Accepted 2020 May 27
Copyright: © 2020 Dharmaratnam et al.
Copyright year: 2020
Copyright holder: Dharmaratnam et al.
License: This is an open access article distributed under the terms of the Creative Commons Attribution License, which permits unrestricted use, distribution, reproduction and adaptation in any medium and for any purpose provided that it is properly attributed. For attribution, the original author(s), title, publication source (PeerJ) and either DOI or URL of the article must be cited.
License URL: https://creativecommons.org/licenses/by/4.0/

Keywords: Goldfish, Cyprinid herpesvirus-2, CyHV-2, Caudal fin, Cell line

Funding: National Surveillance Program of Aquatic Animal Diseases, National Fisheries Development Board, Department of Animal Husbandry, Dairying and Fisheries NFDB/Coord/NBFGR/ 2012-13/16720 dated 11.02.2013 ICAR National Fellow Scheme of ICAR ICAR letter no F.No: Agri.edn.27/2/2015 HRD dated 13.02.2017 Department of Agricultural Research and Education, Ministry of Agriculture and Farmer’s Welfare, Government of India This research was carried out under the National Surveillance Program of Aquatic Animal Diseases, National Fisheries Development Board, Department of Animal Husbandry, Dairying and Fisheries (Grant Number: NFDB/Coord/NBFGR/ 2012-13/16720 dated 11.02.2013) and ICAR National Fellow Scheme of ICAR (ICAR letter no F.No Agri.edn.27/2/2015 HRD dated 13.02.2017), Department of Agricultural Research and Education, Ministry of Agriculture and Farmer’s Welfare, Government of India. The funders had no role in study design, data collection and analysis, decision to publish, or preparation of the manuscript.

==============================
Background

Herpesviral hematopoietic necrosis disease, caused by cyprinid herpesvirus-2 (CyHV-2), is responsible for massive mortalities in the aquaculture of goldfish, Carassius auratus. Permissive cell lines for the isolation and propagation of CyHV-2 have been established from various goldfish tissues by sacrificing the fish. Here, we report the development of a cell line, FtGF (Fantail Goldfish Fin), from caudal fin of goldfish using non-lethal sampling. We also describe a simple protocol for successful establishment and characterization of a permissive cell line through explant method and continuous propagation of CyHV-2 with high viral titer using this cell line.

Methods

Caudal fin tissue samples were collected from goldfish without killing the fish. Cell culture of goldfish caudal fin cells was carried out using Leibovitz’s L-15 (L-15) medium containing 20% FBS and 1X concentration of antibiotic antimycotic solution, incubated at 28 °C. Cells were characterized and origin of the cells was confirmed by sequencing fragments of the 16S rRNA and COI genes. CyHV-2 was grown in the FtGF cells and passaged continuously 20 times. The infectivity of the CyHV-2 isolated using FtGF cells was confirmed by experimental infection of naïve goldfish.

Results

The cell line has been passaged up to 56 times in L-15 with 10% FBS. Karyotyping of FtGF cells at 30th, 40th and 56th passage indicated that modal chromosome number was 2n = 104. Species authentication of FtGF was performed by sequencing of the 16S rRNA and COI genes. The cell line was used for continuous propagation of CyHV-2 over 20 passages with high viral titer of 107.8±0.26 TCID50/mL. Following inoculation of CyHV-2 positive tissue homogenate, FtGF cells showed cytopathic effect by 2nd day post-inoculation (dpi) and complete destruction of cells was observed by the 10th dpi. An experimental infection of naïve goldfish using supernatant from infected FtGF cells caused 100% mortality and CyHV-2 infection in the challenged fish was confirmed by the amplification of DNA polymerase gene, histopathology and transmission electron microscopy. These findings provide confirmation that the FtGF cell line is highly permissive to the propagation of CyHV-2.

Introduction

Cyprinid herpesvirus-2 (CyHV-2), the etiological agent of herpesviral hematopoietic necrosis (HVHN) disease, is a pathogen of goldfish that has been associated with disease outbreaks in goldfish culture (Jeffery et al., 2007). The pathogen first reported as the cause of outbreaks in juvenile goldfish in 1992 from Japan (Jung & Miyazaki, 1995), has subsequently been reported from more than 13 countries (Adamek et al., 2018). In India, CyHV-2 infection was associated with large-scale mortalities of goldfish in West Bengal (Sahoo et al., 2016). As the surviving fish become lifelong carriers of latent infection without exhibiting any clinical signs of the disease (Wang et al., 2012) and goldfish are one of the most traded ornamental fish globally, the international trade of such apparently healthy but latently infected goldfish is suspected to be responsible for the global spread of CyHV-2 (Ito et al., 2013; Adamek et al., 2018).

Cell lines are considered the gold standard for isolation and identification of viruses (Landry, 2009). Earlier, a number of cell lines from other fish species, namely Epithelioma papulosum cyprini cells (EPC), fathead minnow cells (FHM) and koi fin cell lines (KF-1) have been used for CyHV-2 cultivation, but these cell lines did not support its continuous propagation beyond 5th passage (Jeffery et al., 2007; Wang et al., 2012). In an earlier study from India, the virus could not be propagated beyond 4th passage, despite producing cytopathic effects in CCKF cell line (Sahoo et al., 2016),. It has been reported that cell lines derived from homologous species provide sensitive tools for virus propagation (Fryer & Lannan, 1994) and successful cultivation of CyHV-2 has been reported in cell lines developed from goldfish, namely GFTF (Yan, Nie & Lu, 2011), GFF and SRTF (Ito et al., 2013) and GiCB (Ma et al., 2015). More than 15 cell lines have been developed globally from different goldfish tissues since 1964 (Table 1). However, no cell line from goldfish is available in India, therefore, the need of establishing a homologous cell line from goldfish was deemed necessary for the continuous propagation and study of CyHV-2.

Table 1 List of goldfish cell lines.

S. No	Designation of cell line	Tissue	Morphology	Growth medium	Passage level	Chromosome	Revival rate	Viral susceptibility	Country	Reference	
1.	–	Whole fish	–	At 25 °C TC medium 199
with 20% bovine serum	27	–	–	–	United states of America	Kroeker (1964)	
2.	–	Muscle	–	At 25 °C TC 199
with 20% bovine serum	18	–	–	–	United states of America	Kroeker (1964)	
3.	–	Heart	–	At 25 °C TC 199
with 20% bovine serum	18	–	–	–	United states of America	Kroeker (1964)	
4.	SJU-1	Skin tissue	fibroblast-like	At 20 °C Earle’s basal with 20% FBS	110	94	–	IPNV	United states of America	Rio et al. (1973)	
5.	GFSk-S1	Skin	fibroblastic	At 25 °C in L-15 with 10% FBS	over a 1-year period	–	–	–	Canada	Lee, Caldwell & Gibbons (1997)	
6.	GAKS	Scales	epithelial type	DM 160 with
10% FBS	–	–	–	–	Japan	Akimoto, Takaoka & Sorimachi (2000)	
7.	GFF	Fin	–	–	–	–	–	CyHV-2	China	Li & Fukuda (2003)	
8.	GFM	Muscle	epithelial	At 30 °C in L-15 medium with 20% FBS	35	104	92%	IHNV, SHRV,
SVCV, CCV and PNV	United States of America	Rougée et al. (2007)	
9.	GFSB	Swim bladder	fibroblastic	At 30 °C in L-15 with 20% FBS	35	104	93%	–	United States of America	Rougée et al. (2007)	
10.	GFTF	Tail fin	fibroblast-like	At 25 °C in L-15 with 20% FBS	50	104	90%	SHRV,
SVCV and CCV	China	Yan, Nie & Lu (2011)	
11.	SRTF	Fin of standard Ryukin Takafumi	–	At 25 °C MEM
with 10% FBS	–	–	–	CyHV-2	Japan	Ito et al. (2013)	
12.	RKF	Fin of Ryukin	–	At 25 °C MEM
with 10% FBS	–	–	–	CyHV-2	Japan	Ito et al. (2013)	
13.	GFSe	Snout tissue	epithelial-like
cells	At 25 °C in M199
with 15–20% FBS	80	100	–	SVCV, GCRV, IPNV-SP, and CyHV-2	China	Jing et al. (2016)	
14.	GFKf	Kidney	fibroblast-like cells	At 25 °C in M199
with 15–20 % FBS	80	100	–	SVCV, GCRV, IPNV-SP, and CyHV-2	China	Jing et al. (2016)	
15.	GH	Heart	Fibroblast-like and epithelioid cells	At 25 °C in M199
with 10–20% FBS	95	100	90–95%	EHNV, ADIV and BIV	China	Jing et al. (2017)	
16.	GFB	Brain	fibroblastic	At 30 °C in199
with 20%f FBS	40	110	90%	CyHV-2	China	Xu et al. (2019)	
Note:

IHNV, Infectious hematopoietic necrosis virus; SHRV, snakehead rhabdoviruses; SVCV, spring viraemia carp virus; CCV, channel catfish virus; IPNV, infectious pancreatic necrosis virus; CyHV-2, cyprinid herpesvirus 2; EHNV, epizootic hematopoietic necrosis virus; ADIV, Andrias davidianus iridovirus; BIV, Bohle iridovirus; L-15, Leibovitz-15 medium; FBS, foetal bovine serum.

In this article, we report the development of a cell line from the caudal fin of fantail goldfish (FtGF). The cell line has been successfully employed for propagation of CyHV-2, and experimental reproduction of HVHN using supernatant from the CyHV-2 infected cell line (at 10th passage). The newly developed FtGF cell line will play significant role in future research on CyHV-2, including the development of strategies for the prevention and control of the disease caused by CyHV-2 in India.

Materials and Methods

Generation of the primary cell culture

A healthy fantail goldfish weighing 20 g was purchased from a local commercial aquarium shop and kept in properly aerated and filtered fish tank. The fish was fed daily and water changes were performed on alternate days. The goldfish was anesthetized using 3- aminobenzoic acid ethyl ester methanesulphonate (MS-222, Sigma-Aldrich, St. Louis, MO, USA) at the dose rate of 150 mg/L of water and the caudal fin was excised after wiping with 70% ethanol. The excised tissue was washed three times in Phosphate Buffer Saline (PBS) (Life Technologies, Carlsbad, CA, USA; Lot number—1967526; Catalogue number—14190-144 500 mL) containing antibiotic–antimycotic solution (200 IU/mL penicillin, 200 μg/mL streptomycin and 0·5 μg/mL amphotericin B) (Life Technologies, Carlsbad, CA, USA; Lot number—2112695; Catalogue number—15240-062 100 mL). The fin tissue was mechanically cut with a fine scalpel into smaller pieces and then seeded in a 25 cm2 flask. The PBS was removed and attachment of the tissues to the surface was facilitated by adding 200 µl of fetal bovine serum (FBS) (Life Technologies, Carlsbad, CA, USA; Lot number—42G829K; Catalogue number – 10270-106 500 mL). After 2 h, about 7 mL of Leibovitz’s L-15 (L-15) medium (Life Technologies, Carlsbad, CA; Lot number—2085192; Catalogue number—11415-064 500 mL) containing 20% FBS and 1× concentration of antibiotic antimycotic solution was added to the flask, which was then incubated at 28 °C. The flask was observed daily under an inverted light microscope (Nikon Corporation, Tokyo, Japan).

Subculture and maintenance

After formation of monolayer, the cells were washed with PBS followed by trypsinization with 0.25% trypsin EDTA solution (Life Technologies, Carlsbad, CA, USA; Lot number—2053183; Catalogue number—15400-054 100 mL). The cells were observed under an inverted light microscope. The trypsin EDTA solution was discarded immediately on observing cell detachment. The flasks were gently tapped to release the cells from the surface and L-15 medium with 20% FBS was added to stop the action of trypsin. The cells were subcultured at a split ratio of 1:2. The flasks were observed regularly and the cells were subcultured after attaining 80–90% confluence. After 10 subcultures, the concentration of FBS in medium was reduced to 10%.

Cell growth characteristics

Cell growth studies were conducted at different temperatures and FBS concentrations in L-15 medium. The growth rate assays were conducted at temperatures ranging from 20 to 37 °C and FBS concentrations from 5 to 20%. A monolayer flask at 20th passage was trypsinized and the cells were seeded into different six well plates. The plates were incubated overnight at 28 °C. Thereafter, the plates were incubated at different temperatures of 20, 24, 28, 35 and 37 °C and the growth rate was observed. Cells were harvested from the duplicate wells daily and counted with a Neubauer hemocytometer. A similar study was conducted using different six well plates incubated at 28 °C having different FBS concentrations of 5, 7.5, 10, 15 and 20%. The cells at 25th passage at the concentration of 1 × 105 cells per ml were seeded into 24-well plates and cultured for 7 d at 28 °C. The cells were harvested, the number of cells in each well counted with a hemocytometer, and the average value of three wells at each time was used to plot the growth curve. The growth curve experiment of FtGF cells was repeated two times. Cell growth curves were plotted, and the population doubling time (PDT) was calculated based on the growth curve (Weingartl et al., 2002). Growth characteristic experiment assays were carried out for 8 d, and cell number was represented as mean± standard deviation (SD).

Chromosome analysis

The chromosomal counts for FtGF were undertaken from the 30th, 40th and 56th passage flasks. A flask with 80% confluence was treated with one μg/mL colchicine (Sigma, St Louis, MO, USA) for 4 h at 28 °C and then the supernatant was removed, the cells harvested and resuspended in a hypotonic 0.5% KCl solution for 10 min, and then fixed in a 3:1 methanol : acetic acid solution. The cells were pelleted at 367 g at 4 °C for 10 min and washed four times with methanol and acetic acid solution. Smears were prepared following a conventional drop-splash technique (Freshney, 2010), the slides were stained with 10% Giemsa solution and air dried for 30 min. A total of 100 chromosome spreads were observed and counted under a light microscope (Leica, Germany).

Cryopreservation

The monolayers with 80–90% confluency were used for cryopreservation at various intervals of subculturing (7th, 10th, 20th, 30th and 40th). The media was changed one day prior to cryopreservation. Next day, the cells were trypsinised in PBS and centrifuged at 500 g at 4 °C. at a density of 5 × 106 cells/mL, Briefly, the flask was examined using an inverted microscope to monitor the confluence and to check for microbial contamination if any. The spent cell culture medium was removed. The cell monolayer was washed with a balanced PBS salt solution without calcium and magnesium three times and ensuring the adherence of the cells. One ml of 0.25% trypsin EDTA was added to the 25 cm2 surface area flask to cover the monolayer. Excess solution was discarded and the flask was incubated at 37 °C for 4 min. with gentle rocking. The flask was examined for the detachment of cells, the remaining cells were released with gentle tapping and cells were re-suspended in L-15 medium with 20% FBS to deactivate the trypsin. The harvested cells were dispersed by repeated mild pipetting and cells were counted (Freshney, 2010). The cells were suspended in recovery cell-culture freezing medium (Life Technologies, Carlsbad, CA, USA; Lot number—492531; Catalogue number—12648-010 50 mL) aliquoted into two mL sterile cryovials (Nunc), kept at 4 °C for 2 h, −20 °C for 1 h, −80 °C overnight and then transferred into liquid nitrogen (LN2) containers. The cells stored at 20th and 40th passage were revived after 2 months of storage. Briefly, the frozen cells were thawed quickly at 37 °C and added drop-wise to 15 mL complete medium in a centrifuge tube. The cells were centrifuged at 825 g at 4 °C and the pellet was resuspended in seven mL of complete medium. Cell viability was checked with a hemocytometer following trypan blue staining. The revived cells were seeded into 25 cm2 flask and incubated at 28 °C.

Transfection assays

The FtGF cells at the 25th passage (revived from cryopreserved cells) were harvested and grown on coverslip for determining the nature of the cells and to know the transfection efficiency of the cells. The protocols earlier described by Swaminathan et al. (2016b) were followed for immunophenotyping of cells and transfection studies. FtGF cells in 20 fields were counted randomly and the percentage of positive green fluorescent protein (GFP) cells was calculated.

Molecular characterization of the cell line

DNA isolation was carried out by salting out method following Miller, Dykes & Polesky (1988). The DNA was extracted from FtGF cells at 40th passage and muscle tissue of goldfish. PCR was carried out as per Swaminathan et al. (2016b). PCR products of 562 bp and 642 bp were amplified for mitochondrial 16S rRNA and cytochrome c oxidase subunit I (COI) genes using universal primers (F 5′CGC CTG TTT ATC AAA AAC AT 3′ and H 5′CCG GTC TGA ACT CAG ATC ACG T 3′ (Ward et al., 2005) and F 5′TCA ACC AAC CAC AAA GAC ATT GGC AC 3′ and R 5′TAG ACT TCT GGG TGG CCA AAG AAT CA3′ (Palumbi et al., 1991) respectively. The PCR products of both the fragments were sequenced in an ABI 3730 DNA analyzer (Applied Biosystems, Forster City, CA, USA). The sequences of both the mt DNA gene PCR fragments were compared with the published and known sequences in the National Centre for Biotechnology Information (NCBI) database by the basic local alignment search tool (BLAST) (Altschul et al., 1990).

Detection of Mycoplasma contamination in FtGF cell line

Mycoplasma contamination was tested by PCR with 15th and 40th passage FtGF cells grown for 4 days in L-15 medium without antibiotics. Briefly, the harvested cells were centrifuged at 200 g for 10 min and the supernatant was transferred into micro-centrifuge tubes and centrifuged at 250 g to remove debris. The Mycoplasma contamination was checked using EZdetect PCR Kit (HiMedia, Sundara Nagar, Mathikere, Bengaluru) based on amplification of spacer region between 16S rRNA and 23S rRNA. The amplification products were analyzed in 1.5% agarose gel.

CyHV-2 isolation on FtGF cells and viral titer determination

Diseased goldfish obtained from a farm in Kerala, India were confirmed to be infected with CyHV-2 by PCR as per Jeffery et al. (2007). External examination of moribund goldfish revealed clinical signs including simple loss of scales, pale mucous covered gills. Internally, all visceral organs of affected fish were congested and enlarged and white nodules were found on the spleen and the kidney. Gill, kidney and spleen tissues from diseased fish were collected aseptically, and then homogenized in Dulbecco’s Phosphate Buffered Saline (DPBS), freeze-thawed for three cycles and the tissue homogenate was centrifuged at 3,000 g for 30 min at 4 °C. The supernatant was then filtered through a 0.22 µm filter (Millipore) and checked for bacterial contamination before use. Thereafter, 500 µl of the filtered tissue homogenate was inoculated onto 25 cm2 flasks after removing the medium and incubated in a shaking incubator (45 rpm) at 28 °C. In control flasks, maintenance medium was used in place of tissue homogenate. After 1 h adsorption, 6.5 mL of the maintenance medium (L-15 medium with 2% FBS) was added to the FtGF flasks, which were incubated at 28 °C. The inoculated flasks were checked daily for cytopathic effects (CPE). The cells from flasks exhibiting 80–90% CPE were harvested along with supernatant. This cell suspension was frozen at −80 °C for further use. Further, to know the temperature range for CyHV-2 replication, FtGF cells were infected with CyHV-2 and incubated at 18, 20 and 25 °C. For titration, FtGF cells were grown in a 96-well plate with a confluence of 70–80%. After removing the medium, filtered tissue homogenate was diluted ten-fold (10−1–10−9), and 0.1 mL of diluted filtrate was inoculated in triplicate and allowed to adsorb for 1 h. In control wells, DPBS was added in place of filtered tissue homogenate. Thereafter, 0.2 mL of maintenance medium was added to each well and the plate was incubated at 28 °C. The wells were examined daily for the appearance of CPE up to 2 weeks. The virus titer was determined by 50% tissue culture infective dose (TCID50) assay using Reed & Muench (1938) calculations. The viral susceptibility and titration assays of FtGF cells were determined based on three independent experiments.

Experimental challenge studies on goldfish using CyHV-2 propagated in FtGF cells

Clinically healthy goldfish (12–15 cm; 16–23 gm), procured from a local ornamental fish farm, were divided into two groups of 30 fish each and acclimatized in the laboratory aquarium tanks for a week. The tissues from randomly collected goldfish (n = 5) were screened for CyHV-2 using PCR following Jeffery et al. (2007). The fish were anesthetized with MS-222 (Sigma-Aldrich, St. Louis, MO, USA). Fishes in the infected group were challenged with an intraperitoneal (IP) injection of 0.5 mL of 10th passage CyHV-2 FtGF cell culture supernatant, whereas fish in the control group were injected with 0.5 mL maintenance medium. The water temperature was maintained at 28 °C during the experiment and fish were observed daily for clinical signs and mortality. Three goldfish each in the infected and control group were selected randomly after 7 days post-injection (dpi) and screened for CyHV-2 by PCR assay. The experimental challenge study in fish was carried out following ARRIVE guidelines and carried out in accordance with the National Institutes of Health guide for the care and use of Laboratory animals. The experimental challenge trials were evaluated and approved by Institute Animal ethics Committee (IAEC) of ICAR National Bureau of Fish Genetic Resources (NBFGR) vide approval Number G/IAEC/2019/1 dated 04th October 2019.

Confirmation of the CyHV-2 virus

Polymerase chain reaction

The naturally infected tissues from goldfish, supernatant along with cells collected from FtGF flask and infected tissues from experimentally infected goldfish were processed for confirmation by PCR. The harvested FtGF cells were centrifuged and DNA was isolated from cell pellet using DNeasy blood and tissue kit (Qiagen, Hilden, Germany). Concentration and purity of the extracted DNA was determined by measuring OD at 260 and 280 nm using a NanoDrop ND1000 spectrophotometer (Nano Drop Technologies Inc, Wilmington, De, USA). The samples were stored at −20 °C for further analysis. PCR was performed using published oligonucleotide primers CyHVpol-FOR and CyHVpol-REV primers (Jeffery et al., 2007) for confirmation. Briefly, amplification was performed in 25 µl reaction mixture containing 2.5 µl of 10× Taq buffer, 1.5 µl (10 pmol) of each primer, 0.5 µl of dNTPs (2 mM), 0.25 µl of Taq DNA polymerase (5 U µl −1), 1 µl of total DNA and ddH2O to make final volume to 25 µl. The reaction mixture was pre-heated at 95° C for 3 min followed by 40 cycles of 95 °C for 1 min, annealing at 55 °C for 1 min, extension at 72 °C for 1 min and final extension at 72 °C for 10 min. The PCR products were visualized following electrophoresis in 1.5% agarose gel. Representative PCR amplicons from each primer set were purified and sequenced by Sanger sequencing facility (Scigenom Pvt. Ltd, Adyar, Chennai). These sequences of amplified PCR products from all the cases (natural infected goldfish, experimentally infected goldfish and infected FtGF cells) were confirmed by BLAST analysis.

Histopathology

For histopathological examination, tissues including gills, kidney and spleen were collected from moribund fish at 7 dpi and fixed in 10% neutral buffered formalin. Tissues were dehydrated in ascending grades of ethanol, cleared in chloroform and embedded in paraffin wax. Thin tissue sections, 4–5 µm thick were cut and stained with hematoxylin–eosin for examination under a compound microscope.

Transmission electron microscopy

The experimentally infected fish tissues and CyHV-2 infected FtGF cells were fixed with 2.5% glutaraldehyde, post fixed with osmium tetroxide for 1 h at 4 °C, then dehydrated and embedded. A microtome (Leica ultracut UCT) was used to cut thin sections of about 60–70 nm. Sections were stained with uranyl acetate and alkaline lead citrate after mounting on copper grids. Sections were observed and photographed under a Tecnai T12 Spirit transmission electron microscope at 60 kV. Supernatant from CyHV-2 infected FtGF cells displaying CPE were studied by electron microscopy. Subcultured samples were negatively stained with phosphotungstic acid and examined using a High Resolution Transmission Electron Microscope, (FEI-TECNAI-G2 20 TWIN) at 7,800× magnification and an accelerating voltage of 120 kV.

Statistical analysis

In the present study, all statistical analysis of the data was performed with SPSS version 13.0 (IBM, Armonk, NY, USA; www.ibm.com). All results are expressed as mean ± SD. P values < 0·05 were considered statistically significant.

Results

Primary cell culture and maintenance

The primary culture of fin cells from goldfish was achieved using L-15 medium supplemented with 20% FBS at 28 °C. The cells were seen emerging from the sides and edges of caudal fin tissues (Fig. 1A) and they readily attached to the bottom of the flask. A monolayer culture was obtained within 10 days at 28 °C. The monolayer was passaged at 1:2 ratio every 6–7 days. The growth pattern of lag phase (1–3 days), log phase (3–5 days) and stationary phase (5–7 days) was observed. During the initial passages, mixed fibroblast-like and epithelial-like cells were observed and after 10 passages, epithelial cells dominated over fibroblastic cells (Fig. 1B). The FtGF cells have been subcultured for over 55 passages and the cell line has been designated as Fantail Goldfish Fin (FtGF) cell line. The morphology of FtGF cells at 25th passage and at 56th passage are shown in Figs. 1C and 1D.

Figure 1 Photomicrographs of Fantail Goldfish (FtGF) cell line.

(A) Cells emerging from the explant of caudal fin; (B) morphology of FtGF cells at 10th passage; (C) morphology of FtGF cells at 25th passage; (D) morphology of FtGF cells at 56th passage.

Characterization of FtGF cell line

The FtGF cells were characterized to determine the ability to grow at different temperatures and different concentrations of FBS, to check the chromosome number of the cells, to authenticate the species origin by DNA barcoding, to examine for the Mycoplasma contamination, to confirm the nature of the cells by immunofluorescence test and to check the transfection efficiency of the cells. The most favorable conditions for culturing of FtGF cells were tested at different incubation temperatures including 20, 24, 28, 35 and 37 °C as well as at different concentrations of FBS viz., 5, 7.5, 10, 15 and 20%. The cells incubated at 20 and 37 °C showed rounding and started detaching on 2nd day. Cells grew in the flask at 24 °C, but the growth rate was slow. Maximum growth of the cells was observed at 28 °C. At 35 °C, FtGF cells proliferated very fast initially, but became puffy and were dying by the 2nd day. Culture of FtGF cells grew best in L-15 with 20% FBS and a slower growth was recorded with decreasing concentration of FBS. The growth kinetics of FtGF cells did not show much difference in 10, 15 and 20% FBS, but cells in 7.5 and 5% FBS had slower growth. The suitable temperature and FBS concentration for the culture of FtGF cells were found to be 28 °C and 10% FBS, respectively (Figs. 2A and 2B). The growth curve of FtGF cells at passage 25 (Fig. 2C) showed as typical “S” shape and the cells were at latent stage before day 1 after seeding and then the cells proliferated rapidly and went into logarithmic stage from day 1 to day 5. The cell number remained stationary between day 5 and day 8, but began to decline after day 9. The FtGF cells grew and proliferated at a steady rate, and their doubling time was calculated to be 33.9 h at passage 25. The FtGF cell growth was arrested in metaphase using Colchicine at final concentration of one µg/mL. Karyotyping results obtained by counting 100 spreads at metaphase showed 82–114 chromosomes, with a distinct peak for the cell line at 104 diploid chromosomes at 30th, 40th and 56th passage. The majority of the cells (60%) had a diploid chromosome number (2n = 104) at all three passage levels. The diploid karyotype of FtGF cells and its frequency distribution at passage 30 are shown (Figs. 3A and 3B) respectively. The FtGF cells at 20th and 40th passage were revived and these exhibited 70–75% viability. The cells were incubated at 28 °C and a monolayer was established within 15 days. Green fluorescent signals were observed in FtGF cells transfected with 2 μg of pAcGFP1-N1 expression vector and strong green fluorescent signals were detected at 52 h post transfection and transfection efficiency was calculated about 28% thus confirming their potential to be used in gene expression studies. (Fig. 3C). The origin of FtGF cell line was confirmed by partial amplification and sequencing of 16S rRNA and COI genes from FtGF cells and goldfish muscle and comparison with available COI and 16S rRNA sequences in GenBank. Nucleotide sequence analysis revealed 100% similarity with sequences from goldfish muscle and maximum similarity (99%) with COI (Accession number KX145542 and KX145499) and 16S rRNA sequences (Accession number AJ247070, KY231826) of goldfish in GenBank. These data confirm that the origin of the developed FtGF cell line is from goldfish. No target band of spacer region between 16S and 23S rRNA of Mycoplasma was observed indicating that the FtGF cells were free of Mycoplasma contamination.

Figure 2 Mean ± SD (n = 2) growth rate of newly established FtGF cell line.

(A) Growth at different temperatures; (B) growth at different FBS concentration; (C) growth curve of FtGF cell at 25th passage with four phases including latency, exponential growth, and stationary phases. Values plotted are means ± SD of the measurements.

Figure 3 Morphological characteristics and frequency distribution of the chromosomes of the FtGF cell line at passage 30 and transfection efficiency of FtGF cell line.

(A) Phase-contrast photomicrograph of chromosome spread arrested in metaphase (original magnification: ×400); (B) frequency distribution of chromosomes in 100 cells; (C) expression of green fluorescent protein (GFP) in FtGF cells at 25 h passage (200×).

CyHV-2 isolation on FtGF cells and virus titration assays

FtGF cells were infected with CyHV-2 and evaluated by significant CPEs observed. In this study, CyHV-2 was propagated with high viral titer in the FtGF cell line. The FtGF cell line (Fig. 4A) infected with CyHV-2 began to show morphological changes from the 2nd dpi (Fig. 4B) and the CPE such as cell elongation, rounding and cell fusion with cytoplasmic vacuolation were observed from 4th dpi (Fig. 4C) at 28 °C. Cell death started from the 6th dpi (Fig. 4D) and complete detachment was observed by 10 days. The appearance of typical CPE in FtGF cells infected with Indian CyHV-2 was observed at 8 dpi, 6 dpi and 5 dpi when the cells were incubated at 18, 20 and 25 °C respectively after inoculation of the virus, and it was observed that the efficient propagation of Indian CyHV-2 was most rapid (2 dpi) at 28 °C. The infected FtGF cells and the supernatant were confirmed for the presence of CyHV-2 by PCR and sequencing. No CPE was observed in the control cells inoculated with maintenance medium. The TCID50 of the culture suspension harvested from infected FtGF was found to be 107.8 ± 0.26 TCID50/mL (Fig. 5A). The CyHV-2 could be propagated in FtGF flasks for 20 passages.

Figure 4 The cytopathic effect (CPE) in FtGF cells following infection with CyHV-2 at 28 °C.

(A) Uninfected FtGF cells; (B) FtGF cells infected with CyHV-2 at passage 15 at 2 dpi; (C) FtGF cells infected with CyHV-2 at passage 15 at 4 dpi; (D) FtGF cells infected with CyHV-2 at passage 15 at 6 dpi.

Figure 5 Mean ± SD (n = 2) growth curve of CyHV-2 in FtGF cell line and mortality curve of experimentally infected goldfish and gross pathology of infected goldfish.

(A) Growth curve of CyHV-2 (at passage 20) in the FtGF cell line. (B) Cumulative mortality curve of the experimentally infected goldfish (12–15 cm; 16–23 gm), using challenged with 107.8 ± 0.26 TCID50/mL CyHV-2 (passage 10) propagated in FtGF cells. Values plotted are means ±SD of the measurements. (C) Experimentally challenged goldfish showing swollen body and (D) severe inflammation of gills.

Virulence of CyHV-2 propagated in FtGF cells to goldfish

To confirm the infectivity of CyHV-2 propagated on FtGF cell line, an experimental challenge study was carried out in healthy goldfish. The goldfish challenged with 1.4 × 107 TCID50/mL CyHV-2 (passage 10) produced in the FtGF cell line began to die at 5 days post inoculation (dpi) and the mortality reached 100% at 12 dpi (Fig. 5B). The dead fish from infected groups exhibited similar clinical signs seen in naturally infected fish. The clinical signs of challenged goldfish were hemorrhages on the body surface, exophthalmia, pale gills and a swollen abdomen (Figs. 5C and 5D). There was no mortality in the mock-infected group. The tissues from randomly selected fish from the infected groups tested positive for CyHV-2 by PCR and produced CPE in FtGF, while fish from control group were negative. The CyHV-2 recovered from experimentally infected fish was also found to be virulent to goldfish and caused similar symptoms (T. Swaminathan, 2020, unpublished data).

Confirmation of the CyHV-2 virus

The confirmation of the CyHV-2 infection in infected tissues, FtGF cell culture suspension, and experimentally challenged fish was carried out by amplification of fragment of DNA polymerase gene using PCR, histopathological investigation and transmission electron microscopy.

Polymerase chain reaction

Tissue samples (gills, spleen and kidney) collected from naturally infected tissues, FtGF cell culture suspension, and experimentally challenged fish were found to be positive for CyHV-2 in PCR assay. The expected PCR product of 362 bp was obtained in all CyHV-2 positive samples. A GenBank BLAST search on the sequence revealed a high identity to CyHV-2 isolate SYC1 strain (KM200722, 99.9%) and CyHV-2 isolate STJ1 strain (JQ815364, 99.9%). The sequence of the DNA polymerase gene of CyHV-2 isolated from goldfish in this study was deposited in GenBank (GenBank accession nos. KU527548 and KU527549).

Histopathology

Hypertrophied nuclei with marinated chromatin material in sections of gills, spleen and kidney from goldfish experimentally infected with CyHV-2 (Figs. 6A–6C) and inflammatory exudate between the secondary lamellae were observed in sections of gills. The lesions of the experimentally challenged fish were compared with the sections of gills, spleen and kidney in naturally CyHV-2 infected goldfish.

Figure 6 Histopathological lesions in gills, spleen and kidney of a goldfish with Goldfish Herpesviral Hematopoietic Necrosis Disease. Section of goldfish gills.

(A) Gills; (B) spleen and (C) kidney showing margination of chromatin material (arrows).

Transmission electron microscopy

The mature virus particles were observed ultra structurally in cytoplasm of cells in gills as well as spleen tissue of experimentally challenged fish (Figs. 7A and 7B) and CyHV-2 infected FtGF cells that were about 170–180 nm in diameter (Fig. 7C). Supernatant of FtGF cells with CPE were examined by TEM, and viral particles morphologically similar to a herpesvirus were observed (Fig. 7D).

Figure 7 Transmission electron micrograph showing enveloped mature CyHV-2 virions in the ultrathin sections of tissues and infeced FtGF cells.

(A) Gill; (B) spleen cells of experimentally challenged of goldfish and mature virus particles; (C) CyHV-2 infected FtGF cell; (D) purified from CyHV-2 infected FtGF cells.

Discussion

The ease of detecting and isolating the virus using cell lines provides a remarkable advantage in viral disease management. Isolation of virus in cell culture has been considered as the “gold standard” for the viral disease diagnosis for decades. It would be a valuable approach when a viable virus isolate is needed and also to differentiate nonviable virus in a clinical specimen. Over the past few years, a number of fish viruses viz., viral nervous necrosis virus (Azad et al., 2005), iridovirus (George et al., 2015), cyprinid herpesvirus-2 (Sahoo et al., 2016), and carp edema virus (Swaminathan et al., 2016a), have been reported from India and some of these have also been isolated using fish cell lines. As CyHV-2 mainly affects goldfish, development of species-specific cell line will greatly aid in disease health management. Several cell lines have been employed for isolation of CyHV-2 (Rougée et al., 2007; Yan, Nie & Lu, 2011; Ito et al., 2013; Ma et al., 2015; Jing et al., 2016). In India, Sahoo et al. (2016) isolated CyHV-2 using a koi carp cell line, CCKF but could not propagate the virus beyond 4th passage. The continuous culture of CyHV-2 has been challenging due to the lack of permissive cell lines in India. In this article, we have demonstrated the continuous propagation of CyHV-2 using FtGF cell line as well as successful experimental infection of goldfish by intraperitoneal injection with the CyHV-2 propagated in FtGF cells.

The caudal fin has the feature of natural regeneration capacity (Akimenko et al., 2003) accounting for a high in vitro cell proliferation potential (Santos-Ruiz, Santamaria & Becerra, 2005). The FtGF cell line exhibited stable growth over about 55 passages over the past one year. Using the new cell line, we isolated CyHV-2 using filtered tissue homogenate from infected goldfish. Morphological changes in FtGF cells infected with CyHV-2 included cytoplasmic vacuolation, rounding and detachment of cells, as reported previously in other CyHV-2 infected cells. Moreover, CyHV-2 could be continuously propagated over 20 times using FtGF cell line and also consistently produced the similar CPE as the initial passage. Similar to our results, Ma et al. (2015) could also propagate the virus for over 50 passages in GiCB cell line. However, in contrast to our results, Jung & Miyazaki (1995) reported that using epithelioma papulosum cyprini and FHM cell cultures; CyHV-2 could not be sub-cultured beyond the fourth passage. Previously, several attempts have been made to propagate CyHV-2 in different fish cell lines (Li & Fukuda, 2003; Waltzek et al., 2005; Jeffery et al., 2007; Ito et al., 2013; Xu et al., 2013; Jing et al., 2017) but it has not been possible to propagate CyHV-2 beyond 4-6 passages and virus titer from these cells was very low.

In the present study, efficient propagation of CyHV-2 with high viral titer was achieved when the infected FtGF cells was incubated at 28 °C. Jeffery et al. (2007) isolated CyHV-2 from extracts of gill tissue in KF-1 cells at 20 °C, whereas, Xu et al. (2013) observed CPE by CyHV-2 at 10 dpi on the KF cell line when incubated at 25 °C. Similarly, CPE by CyHV-2 was observed at 6 dpi in GFF and SRTF cell cultures (Ito et al., 2013) and GiCB cells (Ma et al., 2015) when incubated at 25 °C.The virus titer was estimated at 107.8 ± 0.26 TCID50/mL, which is much higher than any earlier reported titer for CyHV-2 in other studies. In this study, the removal of cell culture medium from the flask before inoculating the tissue homogenate in FtGF cells, incubation temperature and shaking during adsorption of the viral inoculum, might have increased the viral titer of CyHV-2 in FtGF cell line. Ma et al. (2015) reported that the CyHV-2 titer reached 107.5 ± 0.37 TCID50/mL and has been effectively cultured over 50 times in the GiCB cell line. Ito et al. (2013) reported that incubation temperature of 25 °C is considered to be highly tolerant for isolation of CyHV-2 in GFF cells. But, in our study, we got high CyHV-2 tire of 107.8 ± 0.26 TCID50/mL at 28 °C which is a high incubation temperature. In accordance with our results, Piaskoski, Plumb & Roberts (1999) and McClenahan, Beck & Grizzle (2005) found that incubating the cells inoculated with the largemouth bass virus (LMBV) samples at 30 °C resulted in a higher number of LMBV plaques than incubation at 25 or 32 °C. Similarly, Chi et al. (1999) also found that the optimum temperature range for grouper nervous necrosis virus (GNNV) infection in GF-1 cells was 24–32 °C and virus titer increased with an increase of the temperature. The removal of tissue culture medium might have made the virus concentration higher during adsorption and agitation resulting in a more uniform distribution of the viral inoculum among all the FtGF cells in the flasks. The isolate of CyHV-2 in this study showed a different temperature range for its culture compared to other published reports by various workers discussed earlier. One of the possible explanations could be that the present Indian CyHV-2 might belongs to a different strain or genotype or it could be due to an adaptation of the host fish species to the native environmental temperature (Ciulli et al., 2006).

An experimental infection in goldfish was carried out with the virus passaged 10 times in FtGF cells. In this experiment, fish injected IP with the cell culture grown virus, began to die at 5 dpi and the cumulative mortality reached 100% at 12 dpi. This finding is in accordance with a previous report (Ito et al., 2013), where cumulative mortality of 90 (at 16 dpi) and 100% (at 18 dpi) were observed in Edonishiki and Ryukin goldfish varieties, respectively. In addition, Ma et al. (2015) reported cumulative mortality of 100% at 14 dpi in healthy gibel carp challenged with CyHV-2 (passage 9) produced in the GiCB. The CyHV-2 virus re-isolated from experimentally infected fish in FtGF cells was also pathogenic to goldfish and caused disease in goldfish. In the present study, the electron microscopic observations demonstrated mature virus particles similar to herpesvirus in affected gill and spleen tissues from the experimentally challenged goldfish and these findings are consistent with previous reports (Jeffery et al., 2007; Wu et al., 2013; Ma et al., 2015). CyHV-2 cultured in FtGF passages has been stored in liquid nitrogen. The recovered viruses from liquid nitrogen maintained high infection activity to cells (data not shown). The results indicated that the FtGF cell line is capable of producing high concentrations of CyHV-2 in vitro and a highly permissive to the propagation of CyHV-2.

Conclusion

In conclusion, the newly established FtGF cell line is highly susceptible to CyHV-2 culture in vitro. The newly developed FtGF cell line would play a crucial role in future research on CyHV-2, including studying the molecular pathogenesis of HVHN disease and development of strategies for the prevention and control of the disease in the country. The FtGF cell line (NRFC accession number: NRFC058; http://mail.nbfgr.res.in/nrfc/cellline-available.php) was deposited in the National Repository of Fish Cell Line (NRFC) (the largest fish cell line repository), ICAR National Bureau of Fish Genetic Resources, India for further dissemination to scientists for carrying out research in developing CyHV-2 management strategies in this field.

Ethical statement

All the experimental challenge procedures in this study (Proposal number: NBFGR/IAEC/2019/0014) were evaluated and approved by Institute Animal ethics Committee (IAEC) of ICAR National Bureau of Fish Genetic Resources (NBFGR) (CPCESA Registration No: 909/GO/Re/S/05/CPCSEA dated 09.09.2005 and CPCSEA Ref file No. 25/111/2014-CPCESA dated 05th December 2018) vide approval Number G/IAEC/2019/1 dated 04th October 2019.

Supplemental Information

Supplemental Information 1 Replicates used to calculate means and error bars on the growth requirements of FtGF cells (Figs. 2A and 2B)and karyotype frequecy distribution (Fig. 3B).

Click here for additional data file.

Supplemental Information 2 Data on the cumulative mortality cure in Fig. 5B.

Click here for additional data file.

Supplemental Information 3 Growth curve of CyHV-2 on FtGF cell line for Fig. 5A.

Click here for additional data file.

The authors express their thanks to Director, ICAR-National Bureau of Fish Genetic Resources, Lucknow and Deputy Director General (Fy. Sc.), Indian Council of Agricultural Research (ICAR), New Delhi for their support, guidance and encouragement. Authors thank Rahul G Kumar, PMFGR Centre, ICAR NBFGR for correcting English language of this MS. We thank the Department of Gastroenterology, Christian Medical College, Vellore, India and High Resolution Transmission Electron Microscope Facility, Vellore Institute of Technology, Vellore, India for the TEM analysis of the samples.

Additional Information and Declarations

Competing Interests

Author Contributions

Animal Ethics

Data Availability

The authors declare that they have no competing interests.

Arathi Dharmaratnam performed the experiments, prepared figures and/or tables, and approved the final draft.

Raj Kumar analyzed the data, prepared figures and/or tables, and approved the final draft.

Basheer Saidmuhammed Valaparambil performed the experiments, authored or reviewed drafts of the paper, and approved the final draft.

Neeraj Sood analyzed the data, authored or reviewed drafts of the paper, and approved the final draft.

Pravata Kumar Pradhan analyzed the data, authored or reviewed drafts of the paper, and approved the final draft.

Sweta Das performed the experiments, prepared figures and/or tables, and approved the final draft.

T. Raja Swaminathan conceived and designed the experiments, authored or reviewed drafts of the paper, and approved the final draft.

The following information was supplied relating to ethical approvals (i.e., approving body and any reference numbers):

Fish were not killed or euthanized after the experiment. Only the caudal fin clip was excised following all humane methods under laboratory conditions. The ICAR National Bureau of Fish Genetic Resources Institute Animal Ethics Committee approved this research (approval Number G/IAEC/2019/1 dated 04th October 2019).

The following information was supplied regarding data availability:

Raw data for Figs. 2, 3 and 5 are available as Supplemental Files.

The sequence of the DNA polymerase gene of CyHV-2 isolated from goldfish in this study are available at GenBank: KU527548 and KU527549.

The FtGF cell line is available in the National Repository of Fish Cell Line (NRFC) (the largest fish cell line repository), ICAR National Bureau of Fish Genetic Resources, India: NRFC058 (http://mail.nbfgr.res.in/nrfc/cellline-available.php).

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
