# Peer review of "Establishment and characterization of fantail goldfish fin (FtGF) cell line from goldfish, Carassius auratus for in vitro propagation of Cyprinid herpes virus-2 (CyHV-2)"

_PeerJ, doi:10.7717/peerj.9373_

## Round 0.1 · original submission · Major Revisions

Your paper has to be deeply improved. Important aspects have to be included, such as if cells could be used also to show the temperature range for virus replication. The replicate number has to be added in the graphs description. The statistical analysis has to be done to compare the cell growth speed.

Reviewer 1 ·

Basic reporting

This is a clearly written and technically sound article about a new goldfish cell line in India. There are currently very few cell lines available for HVHN researches. Therefore, every new cell line that is sensitive to CyHV-2 will be warmly welcomed by the scientific community.

Experimental design

Although the introduction gives a good descriptive overview of currently available goldfish cell lines for CyHV-2 propagation, it would be better if the authors could also summarize these data in a short table with names and characteristics of these cell lines and references to the original description in literature.

Materials and Methods section:
Line30:
The manufacturer and lot number of L-15 should be added.
Line31:
The manufacturer and lot number of FBS should be added.
Line 124:
Please describe the trypsinization protocol in detail.

Validity of the findings

Results section:
Fig. 3B:
It looks as if the majority of the cells (60%) did not show a chromosome number of 2n=104 as described in the text, sth wrong with the x-axis markers?

Fig.6:
The scale bar of 6A is totally different with the other three, and I just can’t read the length of the bars of 6B~6D: they are too small and illegible.

Additional comments

Three items would contribute to the value of the paper:
 a quick test to determine if these cells can be transiently transfected with exogenous DNA, such as a lacZ reporter gene plasmid.
 basic Illumina RNAseq analysis of the cell line could provide a good insight into the nature of the cell line.
 Histopathological analysis and transmission electron microscopy results of virus-infected cells are appreciated.

Reviewer 2 ·

Basic reporting

no comment

Experimental design

no comment

Validity of the findings

no comment

Additional comments

Arathi and colleagues describe a simple protocol for successful establishment and characterization of a permissive cell line through explant method and continous propagation of CyHV-2 with high viral titre using this cell line. CyHV-2 is an important virus as it causes high mortality in aquaculture. Cell lines are considered the gold standard for isolation and identification of viruses. The newly developed FtGF cell line will play significant role in future research on CyHV-2. It is a topic of interest to the researchers in the related areas but the paper needs very significant improvement before acceptance for publication. My detailed comments are as follows:
1. The presented MS has so many problems that including mainly language and other basic common errors, and the preparations lacks lot of information in methods and the same reflecting in their results and discussion. Due to the language issues, it is very hard to follow the complete MS. I suggest the authors to rewrite the whole manuscript with suggestions from the qualified scientific writing individuals.
2. Title: seems inappropriate, suggested to change it
3. Introduction: Most of the words seems irrelevant which makes difficult in understand search for similar and appropriate scientific word choices during revision. Some of the sentences are not even at the basic level of technical writing.
4. Materials and methods: In general, tiere is a lack of explanation of replicates and statistical methods used in the study.
5. Results:For every section start with brief introduction of experiment and explain your obtained results in details. You have to provide all your results that are obviously seen in your figures and the figures should be represented that you were describing in this Section.

Reviewer 3 ·

Basic reporting

The Authors clearly explain the steps used to develop the cell-line. This is an important step because growing cyprinid herpesviruses is still a big Challenge in fish research. The back ground is Clear and sufficiently explains the problem faced With Current culture of CyHV-2. The literature cited and References are update.

Tabls and figures used are Clear although it could have been good if the Authors had shown the gross pathology in the challenged fish apart from the histology which they have shown

Experimental design

The experimental design is well explained on how the explants were collected and cultured. They explain the different temperatures used as well as the FBS concentration used. The CPE is Clear in the infected monoloyaer while the presence of virions is well explained by Electron microscopy.

Validity of the findings

the conclusions are well supported by the data provided.

Additional comments

This is an important contribution to the culture of CyHV-2

Reviewer 4 ·

Basic reporting

There are some discrepancies between the graphs and description in the text: the temperatures in which the cells were incubated are different: text gives 15°C, 20°C, 28°C, 30°C, 37°C. fig2b: 20°C, 24°C, 28°C, 35°C, 37°C. Please check and correct.

Experimental design

Authors developed the cells line (characterised it somehow) and used it for propagation of the CyHV-2 with which they were able to infect the fish. This in not how the usability of the cells should be shown: interestingly the cells seems to be highly susceptible to the virus (produce high titor) therefore diagnostic use would be better.

Cell could be used also to show the temperature range for virus replication.

Non of the graphs description presents how many replicates were used.

No statistical analysis was done to compare the cell growth speed.

Validity of the findings

Please see point 2.

Additional comments

The paper has very classical approach to the development of the new cells for culturing CyHV-2. The virus could threaten food security in several countries. In my opinion authors missed the opportunity to use the cells in some useful and innovative way: e.g. comparing the diagnostic value of the cells to the PCR the paper would be definitely worth publishing. The infection with cyprinid herpesviruses is difficult to diagnose using the cell (many cells fails reliably diagnose even the acute infections) and perhaps the new cell line developed by would change this.

---

## Round 0.2 · accepted · Accept

Dear Authors,

I am pleased to confirm that your paper has been accepted for publication in PeerJ.

Thank you for submitting your work to this journal.

Reviewer 4 ·

Basic reporting

No comments

Experimental design

No comments

Validity of the findings

No comments

Additional comments

Authors answered my comment satisfactory. Still diagnostic use of the cells would be better approach especially as Authors performed experimental infection from which well-defined material could be used in estimating diagnostic usability of the cells. I find that this was a missed opportunity from the authors.